# Classification and Properties of Dental Zirconia as Implant Fixtures and Superstructures

**DOI:** 10.3390/ma14174879

**Published:** 2021-08-27

**Authors:** Seiji Ban

**Affiliations:** Department of Dental Materials Science, School of Dentistry, Aichi Gakuin University, Nagoya 464-8650, Japan; sban@g.agu.ac.jp; Tel.: +81-052-751-2561

**Keywords:** dental zirconia, ceramics, implant fixture, implant superstructure, classification, properties, surface modification

## Abstract

Various types of zirconia are widely used for the fabrication of dental implant superstructures and fixtures. Zirconia–alumina composites, such as ATZ and NanoZR, are adequate for implant fixtures because they have excellent mechanical strength in spite of insufficient esthetic properties. On the other hand, yttria-stabilized zirconia has been used for implant superstructures because of sufficient esthetic properties. They are classified to 12 types with yttria content, monochromatic/polychromatic, uniform/hybrid composition, and monolayer/multilayer. Zirconia with a higher yttria content has higher translucency and lower mechanical strength. Fracture strength of superstructures strongly depends on the strength on the occlusal contact region. It suggests that adequate zirconia should be selected as the superstructure crown, depending on whether strength or esthetics is prioritized. Low temperature degradation of zirconia decreases with yttria content, but even 3Y zirconia has a sufficient durability in oral condition. Although zirconia is the hardest dental materials, zirconia restorative rarely subjects the antagonist teeth to occlusal wear when it is mirror polished. Furthermore, zirconia has less bacterial adhesion and better soft tissue adhesion when it is mirror polished. This indicates that zirconia has advantageous for implant superstructures. As implant fixtures, zirconia is required for surface modification to obtain osseointegration to bone. Various surface treatments, such as roughening, surface activation, and coating, has been developed and improved. It is concluded that an adequately selected zirconia is a suitable material as implant superstructures and fixtures because of mechanically, esthetically, and biologically excellent properties.

## 1. Introduction

The number of types of dental zirconia has increased over the past two decades [1,2,3,4,5,6], and it is sometimes difficult to choose the adequate type for each restoration. With the advancement of digital technology, it has become possible to fabricate dental restoratives with high fitting accuracy using CAD/CAM systems [6]. In addition, metal-free restorations are of interest for aesthetic and biological reasons. For example, ceramic implant fixtures have replaced titanium implants and are increasing. Zirconia, which has excellent mechanical, aesthetic, and biological properties, is available as a ceramic implant fixture [7,8,9,10]. Although titanium is generally accepted as a reliable dental implant, zirconia is an alternative material to titanium for metal allergic patients and special patients who pursue aesthetics treatment [11]. It is reported that zirconia implants with modified surfaces result in an osseointegration which is comparable with that of titanium implants [12]. In a comprehensive way, osseointegration is defined as ‘a direct structural and functional connection between ordered, living bone and the surface of a load-bearing implant’ [13,14]. Therefore, zirconia is required for surface modifications to obtain osseointegration to bone [15]. On the other hand, materials for implant superstructures can be broadly classified into resin-based, ceramic, and metal materials, and the number of options is increasing [16,17]. The choice is based on the characteristics of each material whether it is suitable for crowns, abutments, frames, and/or all of them. Dental zirconia has superior properties required for the fabrication of implant superstructures in comparison to other dental materials. In this article, I will introduce the classification and excellent properties of dental zirconia as implant fixtures and superstructures.

## 2. Evolution and Classification of Dental Zirconia

### 2.1. TZP and PSZ

In pure zirconia (ZrO_2_), three crystal phase systems (monoclinic, tetragonal, and cubic) transform with temperature, with the monoclinic phase being stable at room temperature. When zirconia is solidly dissolved in yttrium (Y), calcium (Ca), magnesium (Mg), cerium (Ce), or other ions with a larger ionic radius than that of zirconium (Zr), the tetragonal and cubic phase systems become stable at room temperature [1,2]. When the amount of yttria (Y_2_O_3_) added is over 8 mol%, the cubic phase is stable at room temperature, which is called cubic-stabilized zirconia (CSZ). When yttria is 3 to 8 mol%, tetragonal and cubic phases are mixed at room temperature, and it is called partially stabilized zirconia (PSZ). When yttria is around 3 mol%, the tetragonal phases are close to 100% at room temperature, and it is called tetragonal zirconia polycrystal (TZP), also called toughened zirconia. This yttria 3 mol% tetragonal zirconia polycrystal (3Y-TZP) was an early zirconia that was applied to dentistry as “white metal” [3,4]. Like yttria, Ce-TZP is tetragonally stabilized with adequate amount of ceria (CeO_2_).

PSZ and TZP exhibit a very peculiar phenomenon, i.e., when stress is applied and a crack is formed, the phase transforms from tetragonal to monoclinic near the crack tip, forming a transition zone. This phase transformation is accompanied by a large volume change of about 4%. It is believed that the accumulation of strain energy due to the increase in volume lowers the stress at the crack tip and prevents the crack propagation. This phenomenon is called “stress-induced phase transformation” and is the reason why zirconia has extremely high strength despite being a ceramic.

### 2.2. Zirconia–Alumina Composites

Ce-TZP shows a higher toughness value than 3Y-TZP, but its flexural strength and hardness are low, and it has not been put into practical use until now. To overcome these drawbacks, a composite material in which alumina (Al_2_O_3_) particles were dispersed as the second phase in Ce-TZP was investigated, but sufficient results were not obtained. On the other hand, 3Y-TZP with about 20 vol.% of alumina dispersed has improved flexural strength and fracture toughness, and is supplied as a completely sintered body, which is generally called alumina-toughened zirconia (ATZ) (Figure 1a and Figure 2a). On the contrary, there is alumina in which 3Y-TZP is dispersed by about 20 vol.%, which is called zirconia-toughened alumina (ZTA). In the field of orthopaedics, the 3Y-TZP single-phase material is difficult to accept due to troubles with the 3Y-TZP artificial head [2], and composite materials, such as ATZ and ZTA, are used [18,19,20,21]. ATZ composites have been utilized to dental implant fixtures [22].

Apart from these, the Ce-TZP/Al_2_O_3_ nanocomposite (NanoZR) achieved improved properties based on the concept of mutual nanocomposite [23] (Figure 1b and Figure 2b). This material has an interpenetrated intragranular nanostructure, in which either nanometer-sized Ce-TZP or alumina particles locate within submicron-sized alumina or Ce-TZP grains, respectively. This material design makes it possible to strengthen the 10 mol% ceria- TZP matrix with 30 vol.% alumina. Sub-grain boundaries are formed around the incorporated second-layer particles due to residual stress, and the particle size is virtually reduced. At normal grain boundaries, impurities easily enter and cause a decrease in strength, but at sub-grain boundaries, there are no impurities and the grain boundaries become extremely strong. Therefore, this composite exhibits high fracture toughness and high durability against low temperature degradation [24,25]. Although it has not been used as a crown due to its poor translucency, this composite is expected as an implant fixture, since a method for improving biocompatibility has been established [26].

### 2.3. Evolution of Yttria-Stabilized Dental Zirconia

The first dental application of zirconia was 3Y-TZP (3Y-HA), a conventional zirconia containing 0.25–0.5 wt% alumina, which had high strength but insufficient translucency and was used as a core for veneering ceramics. In 3Y-HA, alumina particles deposit in the boundary of zirconia grain and occur a light scattering (Figure 1c and Figure 2c) [27]. High translucent TZP (3Y) was developed by reducing the alumina content to less than 0.05 wt%, and improved the translucency (Figure 1d and Figure 2d) [28]. By using this zirconia, full zirconia crowns can be formed to the final form using only zirconia without the need for a veneering ceramic, and the color can be partially matched using only internal and/or external stain. In addition, a high translucent PSZ (5Y) with increased Y content was provided, and the translucency was further improved due to the increasing of cubic phase, which is optically isotropic (Figure 1f and Figure 2f) [29].

Since the translucency was further improved, the anterior teeth did not need to be veneered with porcelain, and a monolithic zirconia anterior crown became possible. In this way, the translucency of dental zirconia has been dramatically improved (Figure 3). In 2015, a pre-shade semi-sintered body was provided, making it easier to adjust the color tone. Furthermore, a multi-layer type, in which several layers of pre-shades of the same composition are laminated, has been released [30]. In particular, the highly translucent and polychromatic multilayer type (M5Y) has been released one after another by various companies because it enables the production of aesthetic monolithic zirconia anterior crowns with a minimum of work.

Furthermore, since 2016, polychromatic and hybrid composition multilayer (M3Y-5Y), which are laminates of TZP and PSZ with different compositions and characteristics, have been provided. On the other hand, zirconia using high-strength PSZ (4Y) (Figure 1e and Figure 2e), which is an intermediate composition between high translucent TZP (3Y) and high translucent PSZ (5Y) zirconia, was provided [31]. Polychromatic multilayer type (M4Y) has been also released from 2018 to 2019. It can be considered that the dramatic evolution of yttria-stabilized dental zirconia depends on the stable supply of zirconia sintered powder provided by Tosoh Corporation, Japan (Table 1) [28,29,31].

The zirconia pre-sintered body used in the laboratory is mainly in the form of a disc with a diameter of 98 mm and a thickness of 16 to 35 mm (Figure 4), but the block one is frequently used on the chair side for single crowns to three-unit bridges that supports high-speed firing (Figure 5). These blocks are applicable to the super high-speed sintering such as CEREC system SpeedFire^®^ (Densply Sirona), making it possible to fabricate high-translucent and high-strength prosthetics in a short sintering time.

### 2.4. Classification and Indication of Yttria-Stabilized Dental Zirconia

As mentioned above, various types of yttria-stabilized dental zirconia are marketed one after another by each company (Table 2), and zirconia currently available in all over the world is classified into 12 types (Figure 6). 

When the yttria content is high, there are many cubic phases and the translucency is high, but the strength is inferior because there are few tetragonal phases that contribute to the strength. The translucency of 5Y is improved by 20 to 25% compared to that of 3Y, but the flexural strength is reduced by 40 to 50% (Figure 7). 

Therefore, the higher the translucency, the lower the strength. This contradictory relationship must be fully considered when applying zirconia to dental restorations. In other words, 3Y, which has high strength but low translucency, can be applied to long-span bridges, but is not suitable for anterior tooth crowns. On the other hand, 5Y and M5Y, which have low strength but high translucency, cannot be applied to long-span bridges, but can be applied to anterior tooth crowns and veneers. In the case of the polychromatic and hybrid composition multilayer type, such as M3Y-5Y, the application case of 5Y having low strength is dominant. On the other hand, both 4Y and M4Y are applicable to cases of all sites in sufficient terms of strength and translucency.

In Europe and the United States, there is a strong need to use zirconia as a long-span overdenture, and both 4Y and M4Y are preferred. The intention is also reflected in the product names, e.g., DDcube ONE^®^ by Dental Direkt GmbH and One4All^®^ by Metoxit AG. It implies that both 4Y and M4Y can be applied to any cases of any sites.

## 3. Properties of Dental Zirconia

### 3.1. Physical Properties

Thermal properties of zirconia are extremely specific, and among dental restorative materials, zirconia has extremely low thermal conductivity (alumina, 32 W/mK; quartz, 10.7 W/mK; and zirconia, 2.5 W/mK). The coefficient of thermal expansion in the range of 25 to 500 °C is around 10 × 10^−6^/°C, and there is no significant difference depending on the yttria content. This is because there is no crystal transformation in this temperature range and the coefficient of thermal expansion mainly depends on the strength of the atomic bond. The bulk density of zirconia is about 6, and there is no significant difference depending on the yttria content.

Translucency of zirconia is clearly lower than glass ceramics (Figure 8). In the case of glass ceramics, crystal particles are dispersed in the glass matrix, and the light scattering is small due to the small difference in the refractive index between the glass and the dispersed crystals such as lithium disilicate, leucite, and/or feldspar. On the other hand, 3Y-TZP is a polycrystal of optically anisotropic tetragonal zirconia particles, and light scattering is likely to occur at the grain boundaries. Furthermore, the conventional zirconia (3Y-HA) has a relatively high alumina content (0.25–0.5 wt%), and a small amount of alumina particles are precipitated between the zirconia crystal grains. The light refractive index of alumina is 1.76, which is 18% or more smaller than the light refractive index of zirconia (2.15), and even a small amount of precipitation occurs light scattering. It indicates that the composite materials, such as ATZ and NanoZR, show quite low translucency, since ATZ and NanoZR contain 20 and 30 vol.% of alumina, respectively. Even with the same yttria content, 3Y has a low alumina content (less than 0.05 wt%) and hardly precipitates as alumina particles, so that it has higher light translucency than 3Y-HA. The tetragonal phase accounts for almost 100% of the high-translucent TZP (3Y), and about 30% of cubic phase is mixed in the high-strength PSZ (4Y) with the increase in yttria, resulting in high translucency. Furthermore, the cubic phase increases to about 50% in high-translucent PSZ (5Y) [6]. As the yttria content increases and part of the tetragonal phase becomes cubic phase, which are optically isotropic, the light scattering at the grain boundaries is reduced and translucency is improved.

Zirconia shows higher radio-opacity than titanium and aluminum. Between the two types of zirconia, the radio-opacity of NanoZR is slightly lower than that of 3Y-TZP (Figure 9). This is because NanoZR contains 30 vol.% of alumina and its density is also slightly lower than that of 3Y-TZP. It is concluded that the radio-opacity of dental ceramics and metals strongly depends on photoelectric absorption under the general conditions of dental radiography [30]. On this premise, zirconia exhibits high radio-opacity because of its inherently high-effective atomic number and density, whereby high-contrast radiographic images could be obtained for dental diagnosis when it is used as a dental restorative (Figure 10).

### 3.2. Mechanical Properties

#### 3.2.1. Hardness, Flexural Strength, and Fracture Toughness

The hardness of dental zirconia is almost constant, although their flexural strengths decrease with the yttria contents (Figure 11 and Figure 12). Zirconia material is outstandingly large in both flexural strength and hardness, as compared with other restorative materials.

In addition, the fracture toughness values of 5Y decrease by about 50% from that of 3Y with the cubic phase content due to the increasing of yttria content (Figure 13) [33]. Fracture toughness largely depends on the stress-induced phase transformation as mentioned above and decreases with the content of cubic phase that do not generate this transformation.

The fracture toughness value is an index of impact strength, and a low fracture toughness value means a low impact strength. Therefore, considering that the strength of 5Y dental restoration is inferior to that of 3Y, it is necessary to consider making the 5Y restoration thicker than 3Y. Nevertheless, 5Y has higher flexural strength and higher fracture toughness than glass–ceramic materials such as lithium disilicate [5,6].

#### 3.2.2. Fracture Strength of Zirconia Implant

In general, fracture strength of implant is affected by the implant design (1-piece > 2-piece), material (ATZ > Y-TZP), and abutment preparation (untouched > modified/grinded) [34]. Many comparisons of the strength of titanium implants and zirconia implants have also been reported. Kohal et al. [35] evaluated the fracture strength of the all-ceramic implant–crown systems [one-piece zirconia implants (Y-TZP) restored with Empresss 1^®^ and Procera^®^ crowns] after aging in a chewing simulator in comparison with a conventional titanium implant restored with porcelain-fused-to-metal (PFM). They reported that there was no significant difference between the PFM crown group and the Procera^®^ group. Furthermore, they also reported that the fracture strength of two-piece cylindrical zirconia implants (Y-TZP, restored with zirconia crown veneered a feldspathic porcelain and/or restored with Empress 2^®^ crown) after aging in a chewing simulator in comparison with a conventional titanium implant restored with PFM crown [36]. It was indicated that the fracture strength of two-piece zirconia implant fixture was almost half of the one-piece zirconia implant fixture previously reported. Furthermore, the failure mode during the fracture testing in the zirconia implant groups was a fracture of the implant head and a bending/fracture of the abutment screw in the titanium group.

Furthermore, Chang et al. [37] reported the use of a three-dimensional finite element analysis that the biomechanical parameters of the zirconia implants produced similar performance to titanium implants in terms of displacement, stress on the implant, and bone–implant interface. Therefore, it can be judged that the mechanical properties of the superstructure can be evaluated in the same way for both titanium implants and zirconia implants.

#### 3.2.3. Fracture Strength of Zirconia Superstructure

The authors evaluated the fracture strength of three kinds of monolithic zirconia crown on a titanium abutment [38]. High-translucent TZP (Adamant HT^®^, 3Y), polychromatic multilayer and hybrid-composition zirconia (Adamant Zivino^®^, M3Y-5Y), and high-translucent PSZ (Adamant UT^®^, 5Y) crowns for maxillary premolar were milled from each pre-sintered disc using a CAD/CAM system and fired to produce each crown (Figure 14a–c). Three kinds of monolithic zirconia crown were adhesively cemented with ResiCem^®^ (Shofu) on a titanium abutment (Nobel Biocare), and fixed to an implant replica (Ti6Al4V) with an abutment screw (Ti6Al4V) (Figure 15 left). Three specimens were used for each group.

Furthermore, we evaluated the fracture strength of two lithium disilicate abutments in comparison to zirconia abutment [39,40]. IPS e.max CAD^®^ and IPS e.max Press^®^ (Ivoclar Vivadent) were used for lithium disilicate abutment, and inCoris ZI^®^ (Dentsply Sirona, 3Y-HA) was used for zirconia abutment. Three kinds of ceramic abutment were adhesively cemented with ResiCem^®^ (Shofu) on titanium base (Sirona) for the customized abutment. Each abutment was screwed on the Nobel Biocare external regular platform fixture replica and tightened by torque wrench with 16 N. All ceramic crowns for premolar were fabricated with IPS e.max Press^®^, and adhesively cemented with ResiCem^®^ (Shofu) on the ceramic abutment (Figure 14d–f). Five specimens were used for each group. Specimens were loaded to fracture in a universal testing machine at 0.5 mm/min of cross head speed. Specimens were vertically fixed with a vise, and a vertical force was loaded to fracture in a universal testing machine against the maxillary premolar lingual cusp of the specimen. A polyethylene film, 50 µm in thickness, was installed between the specimen and the load head to avoid an extreme concentrated stress (Figure 15 right).

Among the monolithic zirconia superstructures (a, b, and c), 3Y (a) crown showed the highest fracture strength 3303 N (±660 N), and both M3Y-5Y (b) and 5Y (c) crowns showed nearly the same strength, at 2468 N (±709 N) and 2404 N (±580 N), respectively (Figure 16). All the specimens fractured at the loaded lingual cusp of the zirconia crown (Figure 17a–c). It means that the strength of the crown depends on the material strength of the cusp, indicating the strength of the M3Y-5Y crown was nearly the same to that of the 5Y crown because the composition of cusp of the M3Y-5Y crown is 5Y. In case of the superstructures made with ceramic crown and ceramic abutment (d, e, and f), all the specimens fractured at glass ceramic crowns made of a lithium silicate, and there were no significant differences in the fractured force among the material kind of ceramic abutment (Figure 16). However, the fracture modes were different with the materials for the abutment. Although the fracture origin seems to locate in the veneered IPS e.max Press^®^ for all three kinds of abutment, the abutments made of two kinds of lithium disilicate were entirely broken; on the other hand, the abutments made of zirconia were not damaged (Figure 17d–f).

Rohr et al. [41] also reported that a strong correlation was found between fracture loads and fracture toughness of each ceramic crown material on zirconia implant. Therefore, it is concluded that the fracture strength of superstructures strongly depends on the strength of crowns on the occlusal contact region. It suggests that adequate zirconia should be selected as the superstructure crown, depending on whether strength or aesthetics is prioritized, with due consideration to the occlusal force and repair position of the patient.

Furthermore, El-S’adany et al. [42] determined the fracture strength of three kinds abutment (titanium, zirconia, and alumina) cemented with an IPS e.max Press crown for premolar on titanium implant through titanium screw. They found alumina abutment group was significantly weak. It suggests that zirconia should be employed as implant abutment in the metal-free restoration.

### 3.3. Chemical Properties

#### 3.3.1. Low Temperature Degradation (LTD)

Among the chemical properties, the phenomenon of LTD is strongly dependent on the crystal phase due to the yttria content. LTD is a phenomenon in which the tetragonal phase is transformed to the monoclinic phase when heated in the presence of moisture. It is generally evaluated by an accelerated test at 121 and/or 134 °C, because the change is the largest in 200 to 250 °C and is extremely slow in oral temperature.

The authors reported that LTD of NanoZR at 121 °C is significantly less susceptible than 3Y-HA [25]. It seemed that alumina in NanoZR suppressed the degradation. It has also been reported that alumina grains play a major role in resisting the volume expansion of the zirconia grains associated with LTD [43]. Furthermore, we evaluated LTD of various zirconia at a wide temperature range between 37 and 134 °C until 5 years [44]. Since the tetragonal content decreases as the yttria content increases, not only does the amount of monoclinic phase produced due to LTD decrease, but also the structural stability of the tetragonal phase increases, so LTD is less likely to occur [30,44]. It decreases sharply in 4Y and hardly occurs in 5Y (Figure 18). Therefore, 3Y is the result that LTD is most likely to occur among yttria-stabilized dental zirconia. However, even with the same yttria content, 3Y-HA has a little more alumina and LTD is suppressed.

However, it is judged that even 3Y has sufficient long-term durability in the oral environment if it is mirror polished. The authors reported that when the three types of 3Y were left in physiological saline at 37 °C for 5 years, the average amount of monoclinic phase produced was only 4.8% [44]. This means that there is no strong deterioration in the oral cavity for decades.

#### 3.3.2. Erosion in Water-Based Solution

At intraoral temperatures, zirconia has excellent chemical durability and is rarely eroded by acid and alkali solutions. The authors confirmed that the surface of zirconia (3Y-HA and NanoZR) was highly resistant to inorganic acids such as nitric acids, hydrochloric acid, sulfuric acid, and phosphoric acid at 60 °C [45]. However, after soaking in 6%, 12%, and 24% fluoric acid solutions at room temperature, the surface of zirconia was eroded at higher acid concentrations. The high translucent zirconia (3Y and 5Y) behaved similarly.

In mild environments such as 4% lactic acid (pH 1.9) and 0.1N KOH (pH 12.6) solutions, the surface of zirconia (3Y and 5Y) did not change significantly after 30 days at 60 °C, Glass ceramics and hybrid resins showed serious damage (Figure 19) [44,46,47]. With glass ceramics, the differences in the chemistry between matrix glass and dispersed crystal are evident, with different microstructures displaing different surfaces. IPS e.max CAD^®^ showed selective dissolution of the matrix glass, resulting in a clear edge of a needle-shaped lithium disilicate. Hybrid resins, such as Cerasmart^®^ and Enamic^®^, also show selective dissolution of the filler. In saline solution at 90 °C, the glass ceramics and the hybrid resins showed slightly similar dissolutions, but the zirconia showed no changes [46,47]. When autoclaved at 134 °C, glass ceramics and the hybrid resins showed little dissolution, and the zirconia showed no changes [46].

The presence of glass has a strong influence on the properties of the glass ceramics. For example, the strength of glass decreases when it encounters water, especially under load [48,49]. Our previous studies revealed that the biaxial flexure strength of dental porcelains fused to metal in water is 20% lower than those in air [50,51,52]. Dental porcelain fused to metal include approximately 70 vol.% of feldspathic glass, while pressable lithium disilicate contains 30 vol.% of zirconia-reinforced glass. We determined the biaxial flexural strengths of three pressable lithium disilicates in air and water at room temperature compared with high translucent zirconia (5Y) [53]. The biaxial flexural strength of the three glass ceramics in water was about 20% lower than the value of those in air. However, the biaxial flexural strength of zirconia in water was not as statistically significant as the value in air. Therefore, it is concluded that zirconia has high water resistance, hydrolysis in water hardly occurs at intraoral temperatures, and the strength does not decrease.

It can be concluded that the chemical reactivity of zirconia increases with increasing temperature as mentioned above. However, it reaffirmed that zirconia has excellent chemical durability under the same test conditions compared to other dental restorative materials.

### 3.4. Abrasion Properties

Zirconia is undoubtedly the hardest dental restorative material. However, it can be said that the hardness of the restoratives and the wear of the opposing teeth are irrelevant. In the case of tooth wear, it is summarized as attrition at the occlusal part and toothbrush wear at the non-occlusal part, which correspond to abrasive wear and corrosive wear due to lactic acid of mutans bacteria and chemical action of food and drink. Abrasive wear has a large effect on the opposing restoratives [54]. Abrasive wear includes two-body and three-body wear. In the case of dual abrasive wear, when the restorative is hard and strong and the surface is uneven, i.e., when the surface roughness is large, the tooth wear becomes severe.

In the case of ceramic materials, the surface of the restorative is mirror polished with diamond paste. Glass ceramics have a relatively large surface roughness, whereas a smooth mirror surface is obtained on zirconia. Its surface roughness depends not on the hardness of the material but on the constituent microstructure [55]. Glass ceramics consist of glass matrix and dispersed crystal particles, whereas zirconia is made up of fine homogeneous zirconia polycrystals. There are many reports that occlusal enamel wear loss associated with mirror-polished zirconia was minimal compared to glass ceramics [56,57,58,59]. The authors evaluated the wear behavior by measuring the friction coefficient [60]. Zirconia had a smaller friction coefficient than the hybrid resin and did not change, even if the number of times of wear increased (Figure 20) [61]. Through scanning electron microscope observation, it was confirmed that the zirconia surface did not change even by the sliding operation and the smooth surface was maintained. However, this is an effect limited to a smooth zirconia surface, and if polishing is insufficient, the coefficient of friction is large and increases with the number of times of wear. Insufficient polishing will result in hard and strong zirconia being filed. Therefore, it is necessary to recognize that the zirconia restorative does not easily wear the opposing teeth only when it is mirror polished [62]. Grinding/polishing materials and instruments for zirconia are getting better and better. It can be completed in a short time by a suitable operation. As described above, tooth wear is affected by many factors, but compared to other restorative materials, mirror-polished zirconia can be determined to have the least wear on the opposite teeth.

### 3.5. Biological Properties

#### 3.5.1. Bacterial Adhesion

Bacterial adhesion and plaque accumulation on zirconia have been studied in vitro and in vivo, primarily in comparison to titanium. There are many reports that zirconia has less bacterial adhesion and less biofilm formation than titanium [63,64,65,66]. Nakazato et al. [67] measured the bacterial accumulation of six dental materials and reported that zirconia showed low bacterial accumulation compared to other materials. In the early stages of bacterial adhesion (4 h), surface roughness may play a more important role in bacterial adhesion than the surface energy (surface tension) of the implant surface, but at 48 h, there are differences between the materials. This surface energy (surface tension) varies greatly depending on the reporter. In our measurement of distilled water, zirconia showed the largest contact angle among the dental restoratives and the smallest surface energy [68]. However, this contact angle is greatly affected by the surface roughness, i.e., the finish polishing method, and when the surface is mirror polished and the surface becomes smooth, the contact angle is large (Figure 21) and the surface energy is small [68]. Therefore, bacterial adhesion to zirconia can be reduced by mirror polishing the zirconia surface. It indicates that the surface of zirconia superstructure should be mirror polished with diamond paste.

#### 3.5.2. Adhesion of Soft Tissue

The authors investigated the cell adhesion between mucosal epithelial cells and two types of zirconia (3Y and NanoZR), pure titanium, and alumina to compare the adhesion between the cervical region and the gingiva of the dental restoration [69]. Human oral mucosal epithelial cells (Ca9-22) were cultured on these mirror-polished disc specimens, and the cell morphology, cell activity, and mRNA of integrin β_4_, laminin γ_2_, catenin δ_2_, and E-cadherin, which are the adhesion molecules of epithelial cells. The expression level was examined. No significant difference was observed in the cell morphology, activity, and expression level of the adhesion molecules of these epithelial cells on the surface of zirconia and titanium after 1 and 24 h (Figure 22). As a result, it was concluded that zirconia binds to hemidesmosomes, similar to titanium [69]. Other previous studies also confirmed that zirconia binds to the gingival epithelium in a hemidesmosome [70,71].

Yamano et al. [71] evaluated the morphology and cell proliferation of human gingival fibroblasts cultured on four material surfaces [injection-molded smooth surface zirconia (Zr-S), acid-treated rough surface zirconia (Zr-R, corresponding to ZLA^®^ treatment), grade-4 smooth-surface pure titanium (Ti-S), and sandblasted and acid-treated rough surface pure titanium (Ti-R, corresponding to SLActive^®^-treated)] by MTT assay for 24 h and 72 h. Smooth zirconia (Zr-S) showed the highest value in both elapsed times, and cell proliferation grew significantly faster than other surfaces. In addition, smooth zirconia (Zr-S) spread cells most evenly on the sample surface [71]. Therefore, it can be judged that the epithelial adhesion is improved by mirror finishing the zirconia surface. This behavior is advantageous for zirconia abutment.

## 4. Surface Modification for Better Adhesion of Hard Tissue

Zirconia has high strength, but as mentioned above, its polished surface is hydrophobic and is classified as a bioinert material. However, hydrophilicity is desired as an implant fixture to be bonded to living hard tissue.

Therefore, surface modification is required, but it has not been established yet. Furthermore, it is feared that these treatments may cause a decrease in strength, which is an advantage of zirconia. The increase in toughness of zirconia with sandblasting involves the crystal transformation from tetragonal to monoclinic crystals. When other substances enter the surface layer, this transformation behavior becomes hard to occur and the strength unavoidably decreases. Many methods have been proposed to resolve this contradiction and balance the degree of strength reduction with the improvement of biocompatibility [7,8,9,10,11,12,72,73,74,75,76]. In fact, commercially available zirconia implant systems employ various surface modifications (Table 3). These surface modification of zirconia is roughly divided into three methods: roughening, surface activation, and coating as follows.

### 4.1. Roughening

#### 4.1.1. Sandblasting

Zirconia is hydrophobic when mirror polished, but it can be improved to some extent by surface roughness, such as grinding and sandblasting. The authors examined the MTT assay of MC3T3-E1 cells cultured on pure titanium, alumina, 3Y-HA, and NanoZR plates [77]. The number of cells increased in each material as the number of culture days increased, and no significant difference was observed between the materials. Furthermore, no significant difference was observed in cell morphology and actin staining (Figure 23). However, the surface roughness of each material had a large effect, and the rough surface by sandblasting showed significantly better initial cell adhesion. Therefore, zirconia is considered to have the same level of biocompatibility to hard tissue as pure titanium and alumina, and there are many reports like this result [78,79,80].

#### 4.1.2. Etching

Zirconia implant fixtures have rapidly become commercially available, mainly in Europe. Many products are only sandblasted on the surface, but some are acid-treated. Since zirconia has high acid resistance, it is treated with heated hydrofluoric acid [81,82]. Straumann has commercialized this treatment and made it a ZLA^®^ treatment, and has reported that it showed the same osteosynthesis as the SLA^®^ treatment of titanium [83,84].

#### 4.1.3. Laser Irradiation

The effect of laser irradiation on zirconia varies greatly depending on the type of laser. We reported that the surface of zirconia was blackened and cracked by Nd:YAG laser irradiation [85]. CO_2_ and Er:YAG lasers did not cause such damage. However, it has been reported that the CO_2_ laser improves hydrophilicity and improves the adhesion of human osteoblasts (hFOB) [86]. In addition, surface treatment using a femtosecond laser is also being studied. Femtosecond laser processing involves using an ultrashort optical pulse laser with a pulse width of 1 picosecond or less. When an insulator such as zirconia is irradiated, it is thought that the electrons in the valence band transition to the conduction band due to the multiphoton absorption process (dominant over avalanche), the conduction electrons increase, and the behavior becomes similar to that of metal. It can be used for microfabrication and ablation only near the surface. It has been reported that, when a femtosecond laser is applied to a zirconia implant, damage due to laser irradiation is minimized, surface contaminants are reduced, the crystal phase is not changed, and the zirconia implant can be made porous [87,88].

#### 4.1.4. Sintering

A method of forming a porous zirconia layer on the surface by applying a zirconia slurry containing a foaming agent to the surface of calcined zirconia and firing has been reported. It has been reported that implants were made in the femur and cervical bone of rabbits to obtain good binding (removal torque), new bone mass, and connective tissue equivalent to those of Nobel Biocare’s Ti-Unite^®^ [89]. This porous zirconia implant was commercialized as ZiUnite^®^ from Nobel Biocare, and it has been confirmed that it is equivalent to Ti-Unite^®^ in the differentiation of osteoblasts on the surface [90,91]. This implant is not available on the market.

### 4.2. Surface Activation

#### 4.2.1. Ultraviolet Irradiation

Although UV irradiation of titanium is effective, the bioactivation treatment effect of irradiation of zirconia is often negative [92,93].

#### 4.2.2. Plasma Irradiation

It has been reported that it is possible to clean the prosthesis before it is placed in the oral cavity and activate its properties, and it is effective in improving the bond between zirconia and the resin adhesive [94]. By improving hydrophilicity, it also has the effect of improving cell adhesion, but there is a problem with the sustainability of this effect, and, if left in the air, it will return to its original state within a few days [95].

### 4.3. Coating

#### 4.3.1. Glass Coating

Zirconia has properties close to metals among ceramics, and titanium has properties close to ceramics among metals. For example, the coefficient of thermal expansion is close to 10.1 × 10^−6^/K for zirconia and 8.5 × 10^−6^/K for titanium. Therefore, surface activation coating treatment by heating is applied to zirconia, including the glass coating used for titanium. Kim et al. have reported a method of coating the surface of zirconia with fluoroapatite using the sol–gel method [96]. However, it is 1 μm thick and may disappear if it is implanted in the body for a long period of time. Furthermore, Kim et al. [97] have reported a method of baking a mixture of glass and HA on the surface of zirconia. However, glass is sodium–calcium–phosphate-based and is inferior in strength and chemical durability. Further, it is not a laminated structure but only a layer firing and does not consider the concept of matching the coefficient of thermal expansion. On the other hand, Ferraris et al. [98] bake bioactive glass with a high calcium content on the surface of zirconia, and report that calcium phosphate is deposited on the surface when it contacts with body fluids, improving bioactivity.

#### 4.3.2. Apatite-Containing Glass Coating

The authors used silica SiO_2_, borax Na_2_B_4_O_7_ 10H_2_O, and tricalcium phosphate Ca_3_(PO_4_)_2_ as the basic composition of glass and dissolved and pulverized to prepare the glass powder. In the composition containing a large amount of tricalcium phosphate, hydroxyapatite is precipitated in the glass. This glass was made into a paste and applied to zirconia and fired to form a film. The adhesive strength between glass and zirconia depends on the borax content [99]. The amount of osteocalcin produced on the surface of zirconia under coating with this glass was significantly higher than that on the surface without coating [100].

#### 4.3.3. Apatite Coating

Phosphate compounds are highly reactive with zirconia, and when heated in contact with them, they react with yttrium added to stabilize zirconia to produce yttrium phosphate [101]. As a result, the stabilizing element yttrium is reduced from the zirconia, the crystal phase is transformed into monoclinic crystals, and the strength is reduced [102]. Therefore, it is difficult to directly melt and bond the calcium phosphate salt to the surface of zirconia by heating. Uchida et al. [103] reported that apatite was deposited on the surface of zirconia by immersing it in a simulated body fluid after treatment with alkali or acid. However, the bond between the precipitated apatite and zirconia is weak. The authors devised a method to deposit calcium ions on the surface of zirconia by coating and firing, and to deposit apatite on the surface by immersing it in a phosphoric acid-containing solution. Good proliferation of osteoblast-like cells (MC3T3-E1) was observed [104].

## 5. Conclusions

The present review classified the various dental zirconia and summarized their properties for implant superstructures and fixtures as follows:Dental zirconia continues to increase and is classified into 12 species in the yttria system alone. They are classified with yttria content, monochromatic/polychromatic, uniform/hybrid composition, and monolayer/multilayer.Higher yttria content zirconia has higher translucency and lower mechanical strength. Fracture strength of superstructures strongly depends on the strength on the occlusal contact region. Therefore, adequate zirconia should be selected as the superstructure crown, depending on whether strength or esthetics is prioritized.Low temperature degradation of dental zirconia decreased with yttria content, but even 3Y zirconia has a sufficient durability in oral condition.Although zirconia is the hardest dental materials, zirconia restoratives rarely wear the antagonist teeth when it is mirror polished.Zirconia has less bacterial adhesion and better soft tissue adhesion when it is mirror polished. Therefore, zirconia has advantageous for implant superstructures.Zirconia–alumina composites, such as ATZ and NanoZR, are adequate for implant fixtures because they have excellent mechanical strength in spite of insufficient esthetic properties.Zirconia is required for surface modification to obtain osseointegration to bone. Various surface treatments, such as roughening, surface activation, and coating, have been developed and improved.

Therefore, it is concluded that an adequately selected zirconia is a suitable material as an implant superstructure and fixture because of its mechanically, esthetically, and biologically excellent properties.

## Figures and Tables

**Figure 1 materials-14-04879-f001:**
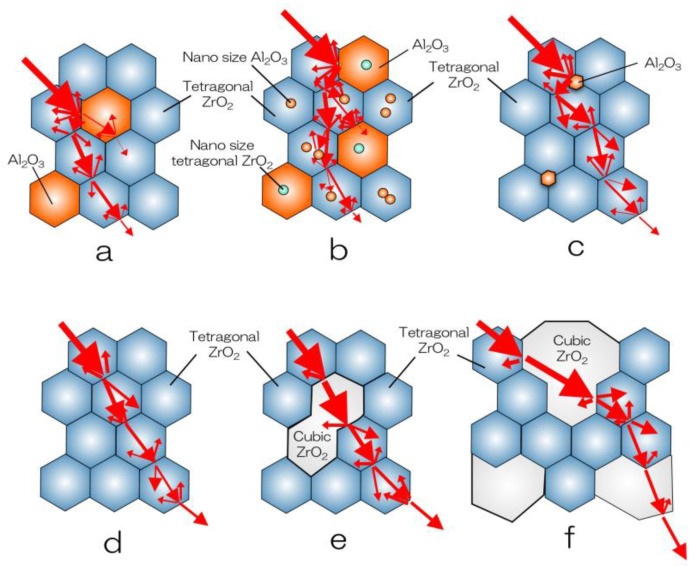
Schematic illustration of microstructure and light scattering of dental zirconia. (**a**) alumina-toughened zirconia (ATZ); (**b**) Ce-TZP/alumina nanocomposite (NanoZR); (**c**) conventional TZP (3Y-HA); (**d**) high translucent TZP (3Y); (**e**) high strength PSZ (4Y); and (**f**) high translucent PSZ (5Y).

**Figure 2 materials-14-04879-f002:**
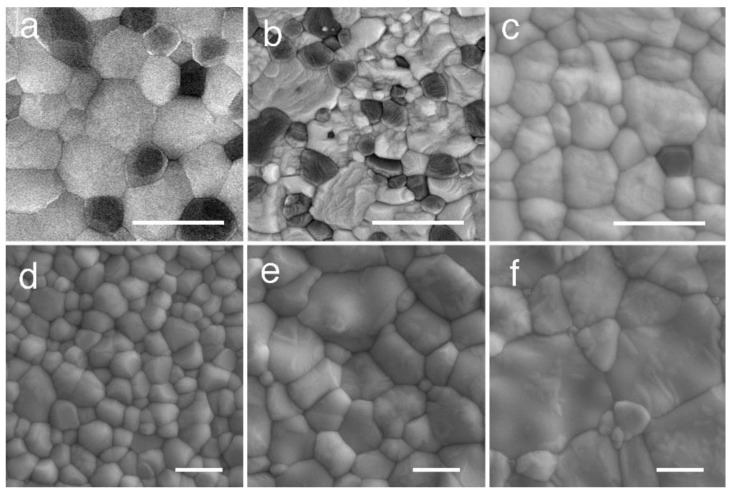
SEM photograph of dental zirconia illustrated in Figure 1. (**a**) alumina-toughened zirconia (ATZ); (**b**) Ce-TZP/alumina nanocomposite (NanoZR); (**c**) conventional TZP (3Y-HA); (**d**) high translucent TZP (3Y); (**e**) high strength PSZ (4Y); and (**f**) high translucent PSZ (5Y). White lines indicate 1 µm. Adopted from Refs. [3,5].

**Figure 3 materials-14-04879-f003:**
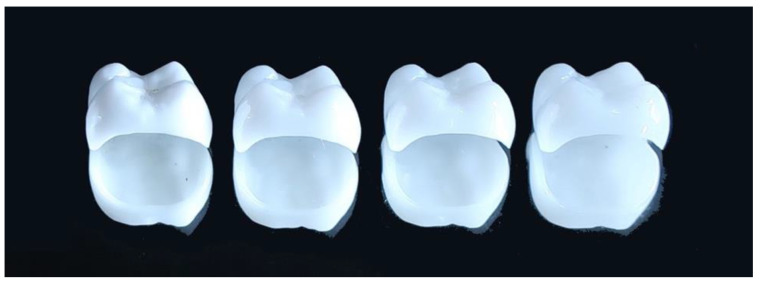
Monolithic zirconia posterior crowns. From the left, 3Y-HA, 3Y, 4Y, and 5Y zirconia. Translucency increases with yttria content.

**Figure 4 materials-14-04879-f004:**
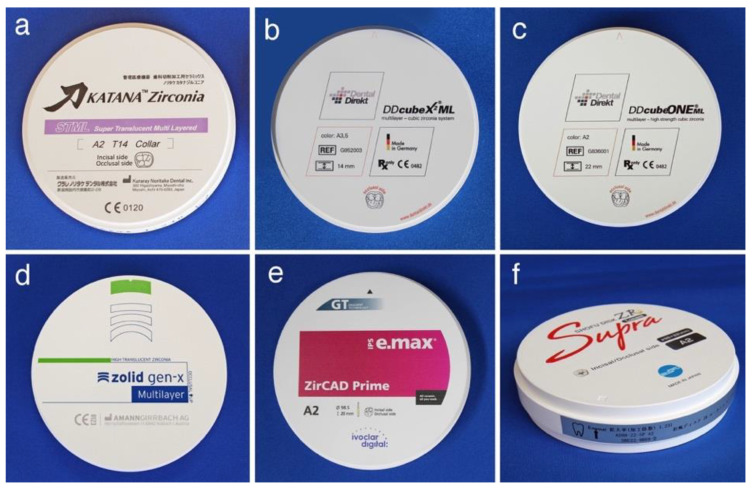
Zirconia CAD/CAM discs. (**a**) Katana^®^ Zirconia STML (Kuraray Noritake Dental, M5Y); (**b**) DDcube X^2®^ ML (Dental Direkt, M5Y); (**c**) DDcube ONE^®^ ML (Dental Direkt, M4Y); (**d**) zolid gen-x^®^ multilayer (Amann Girrbach, M4Y); (**e**) IPS e.max ZirCAD^®^ Prime (Ivoclar Vivadent, M3Y-5Y); and (**f**) Shofu Block Zr Lucent^®^ Supra (Shofu, M3Y-5Y).

**Figure 5 materials-14-04879-f005:**
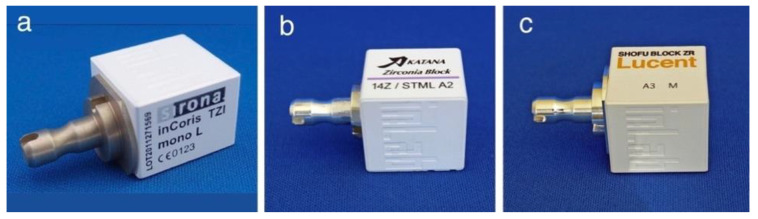
Zirconia CAD/CAM blocks for SpeedFire^®^. (**a**) inCoris TZI^®^ (Dentsply Sirona, 3Y); (**b**) Katana^®^ Zirconia Block STML (Kuraray Noritake Dental Inc., M5Y); and (**c**) Shofu Block Zr Lucent^®^ CEREC (Shofu Inc., M4Y).

**Figure 6 materials-14-04879-f006:**
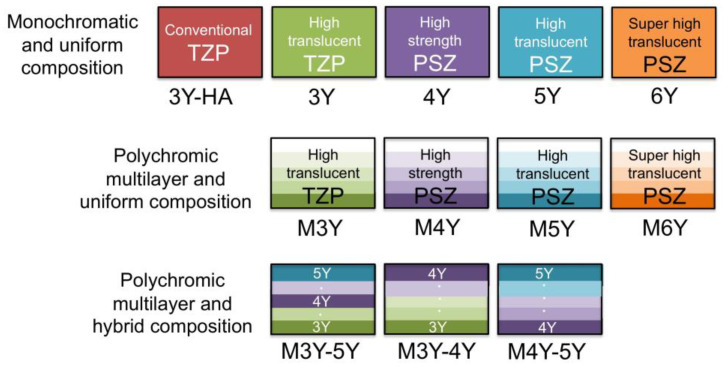
Structural schematic diagram and classification of yttria-stabilized dental zirconia.

**Figure 7 materials-14-04879-f007:**
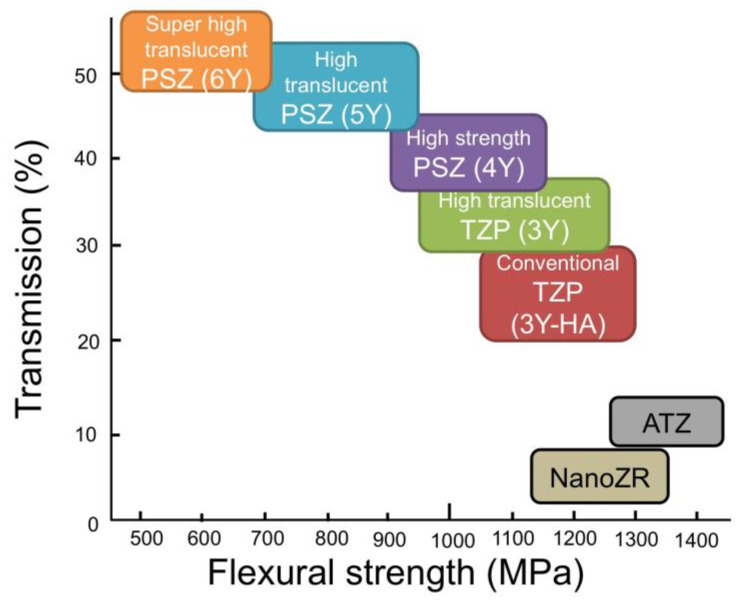
Relation between flexural strength and translucency of classified dental zirconia.

**Figure 8 materials-14-04879-f008:**
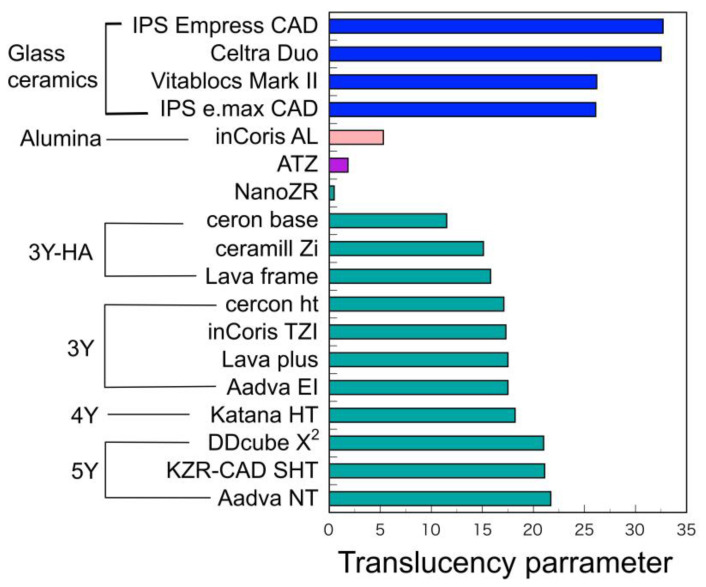
Translucency parameter of CAD/CAM materials. Adopted from manufacturers indication.

**Figure 9 materials-14-04879-f009:**
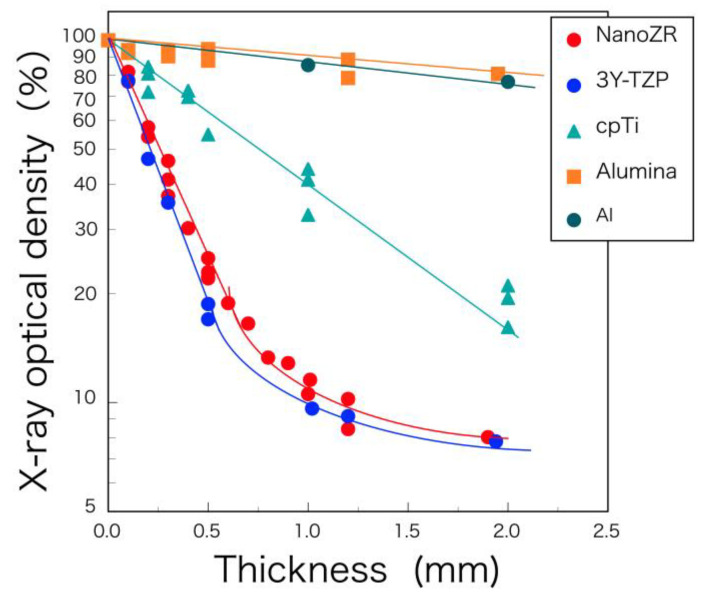
Logarithm of X-ray optical density of NanoZR, 3Y-TZP, cpTi (commercially pure titanium), alumina, and aluminum as a function of thickness. Reprinted with permission from Ref. [32]. 2010 Yuji Okuda.

**Figure 10 materials-14-04879-f010:**
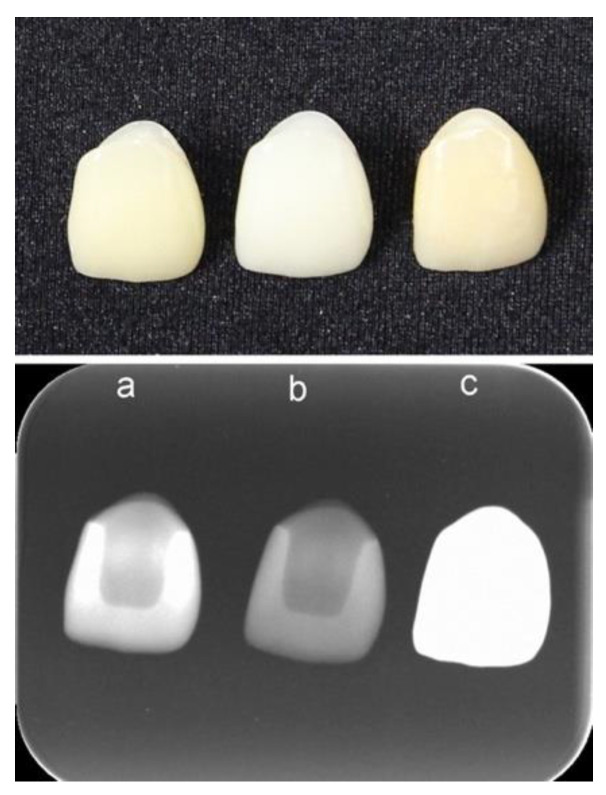
Photograph (upper) and X-ray image (lower) of anterior crowns. (**a**) CAD/CAM hybrid resin (GC, Cerasmart^®^); (**b**) pressed lithium disilicate (GC, LiSi^®^ Press); and (**c**) 5Y zirconia (Shofu Lucent^®^ FA).

**Figure 11 materials-14-04879-f011:**
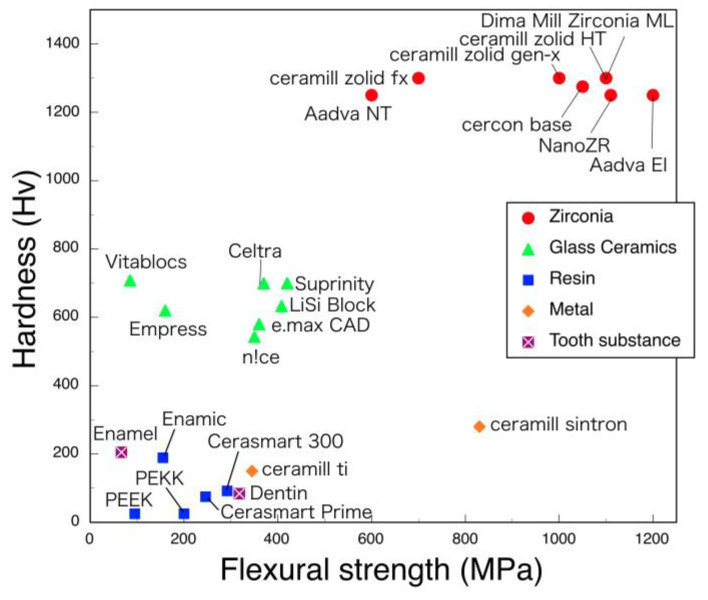
Relation between flexural strength and hardness of tooth substance and CAD/CAM materials. Adopted from manufacturers indication.

**Figure 12 materials-14-04879-f012:**
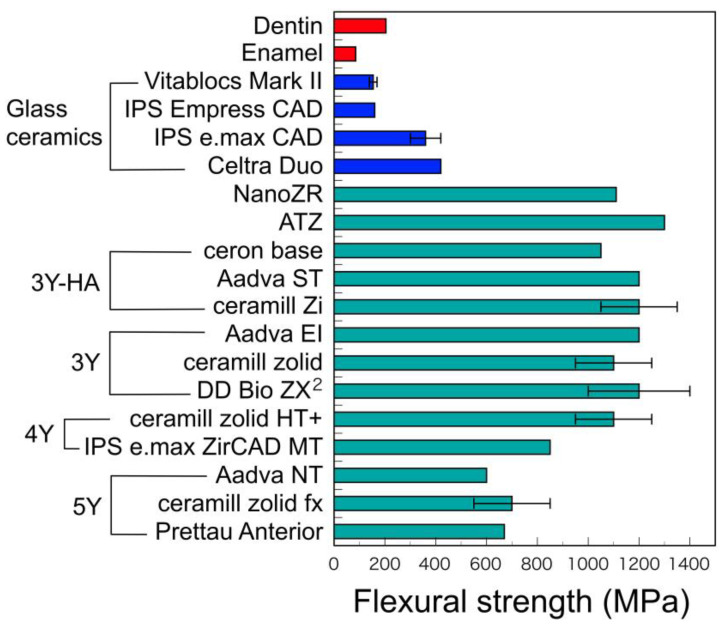
Flexural strength of tooth substance and CAD/CAM materials. Adopted from manufacturers indication.

**Figure 13 materials-14-04879-f013:**
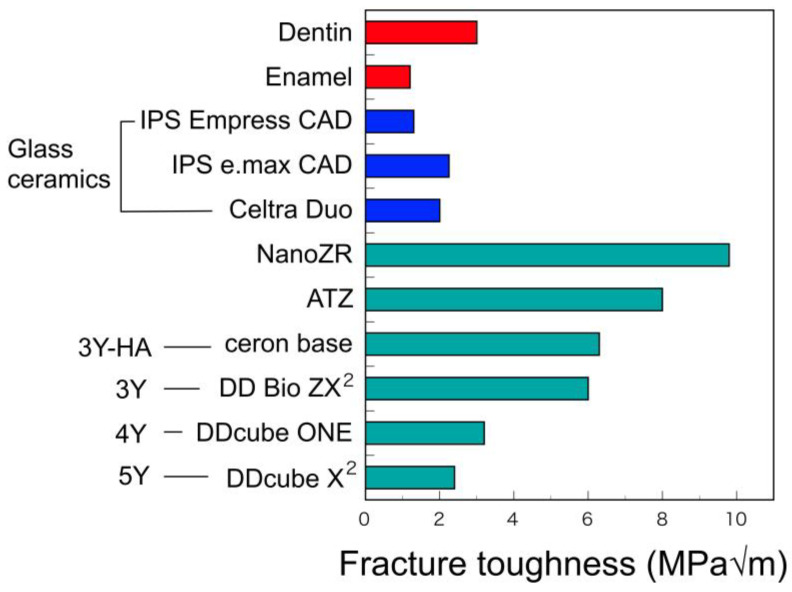
Fracture toughness of tooth substance and CAD/CAM materials. Adopted from manufacturers indications and Ref. [33].

**Figure 14 materials-14-04879-f014:**
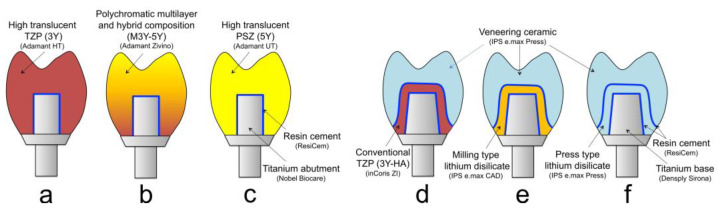
Illustration of fracture test specimens for the superstructure of monolithic zirconia crown (**a**–**c**) and ceramic abutment veneered with lithium disilicate (**d**–**f**). Adopted from Refs. [38,39,40].

**Figure 15 materials-14-04879-f015:**
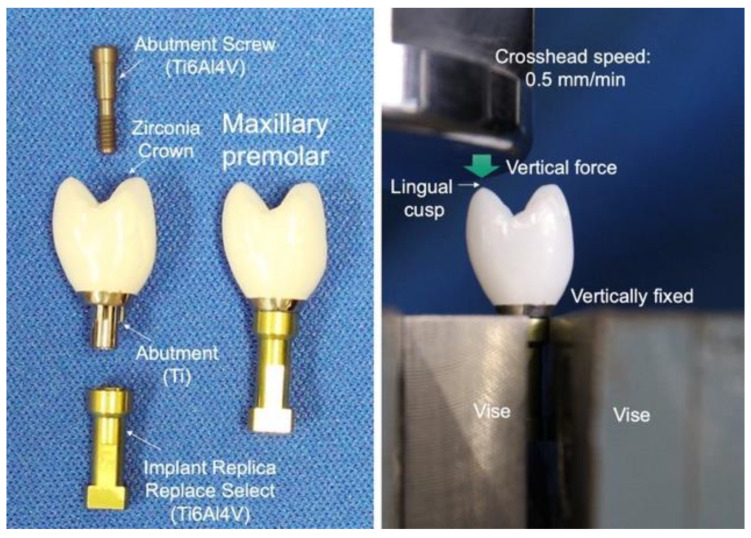
Fracture test specimen of monolithic zirconia crown on a titanium abutment (**left**) and test configuration (**right**).

**Figure 16 materials-14-04879-f016:**
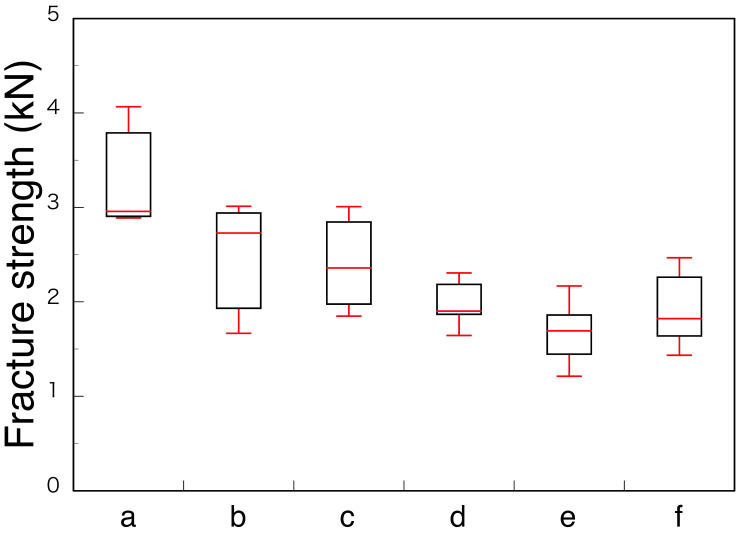
Box plot of fracture test results for the superstructure of monolithic zirconia crown (**a**–**c**) and ceramic abutment veneered with lithium disilicate (**d**–**f**). Adopted from Refs. [38,39,40].

**Figure 17 materials-14-04879-f017:**
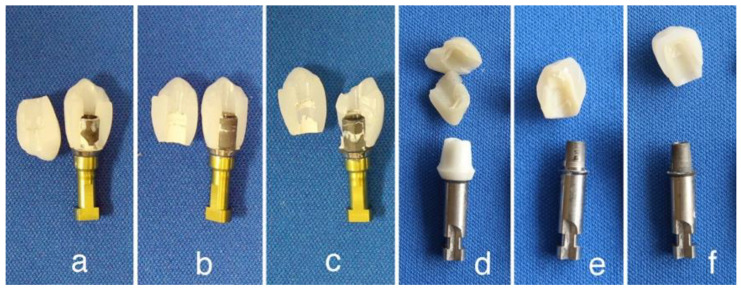
Photographs of fracture specimens for the superstructure of monolithic zirconia crown (**a**–**c**) and ceramic abutment veneered with lithium disilicate (**d**–**f**). Adopted from Refs. [38,39,40].

**Figure 18 materials-14-04879-f018:**
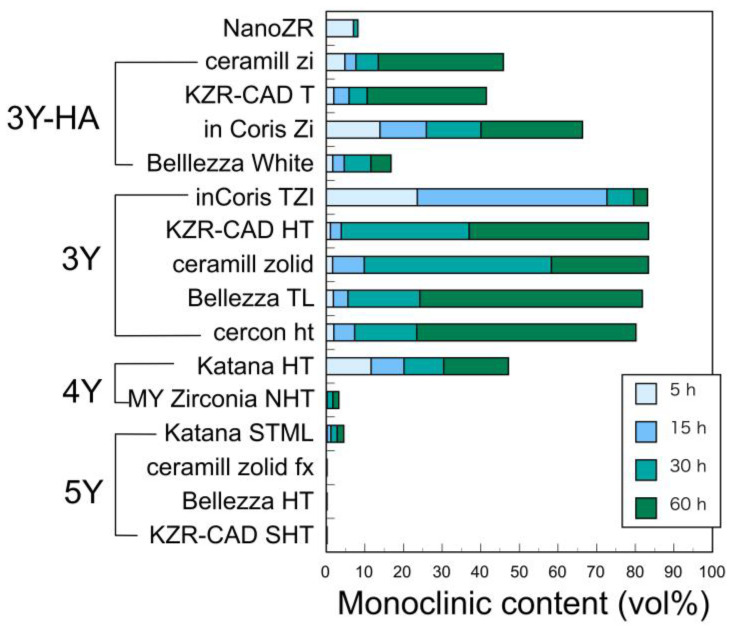
Monoclinic content of dental zirconia after storage in autoclave at 134 °C for 5–60 h. Adopted from Ref. [44].

**Figure 19 materials-14-04879-f019:**
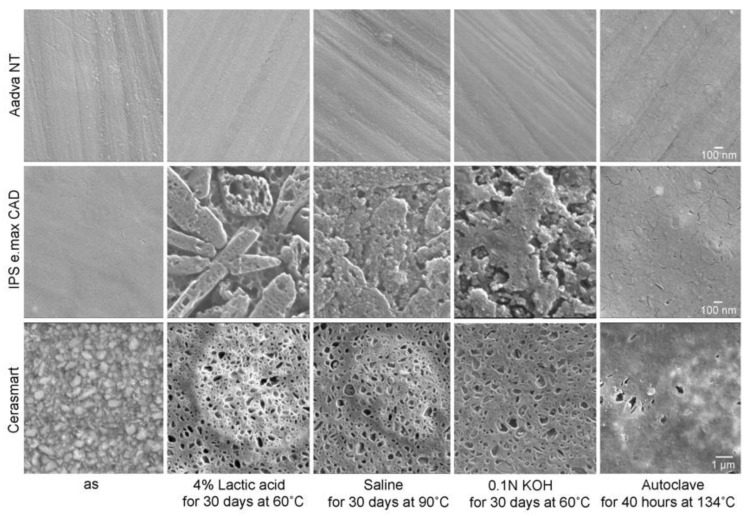
Secondary electron images of high translucent zirconia (5Y) (Aadva NT^®^), lithium disilicate (IPS e.max CAD^®^), and hybrid resin (Cerasmart^®^) before and after soaking in 4% lactic acid for 30 days at 60 °C, saline solution for 30 days at 90 °C, and 0.1N KOH for 30 days at 60 °C, and after autoclaving for 40 h at 134 °C. Reprinted with permission from Ref. [44]. 2020 Seiji Ban.

**Figure 20 materials-14-04879-f020:**
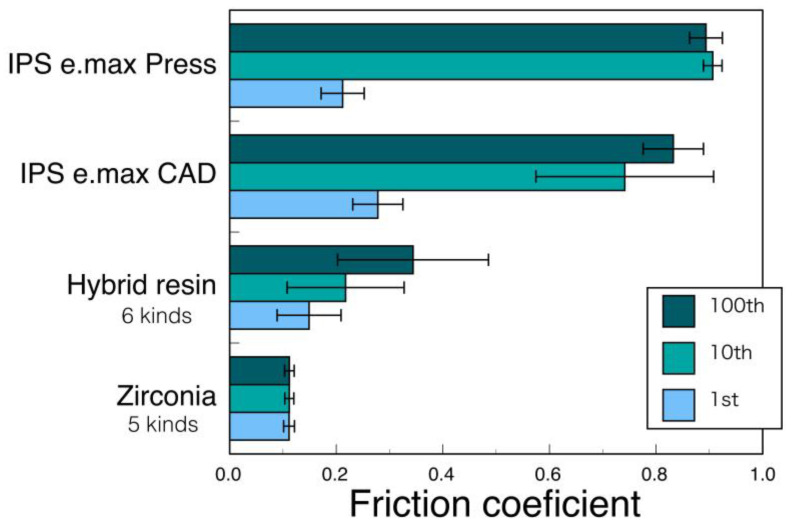
Friction coefficients of two glass ceramics, six kinds of hybrid resin, and five kinds of zirconia against steatite ball in water. Adopted from Ref. [61].

**Figure 21 materials-14-04879-f021:**
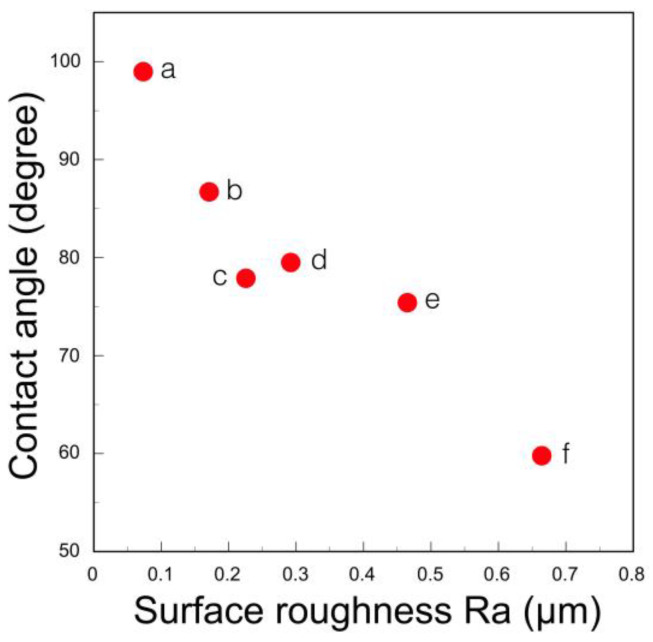
Contact angle of distilled water on surface treated zirconia. (**a**) Millar polish with a diamond paste; (**b**) grinding with a fine diamond bar; (**c**) grinding with a medium diamond bar; (**d**) grinding with #800 abrasive paper and fired; (**e**) grinding with a course diamond bar; and (**f**) sandblast with alumina. Adopted from Ref. [68].

**Figure 22 materials-14-04879-f022:**
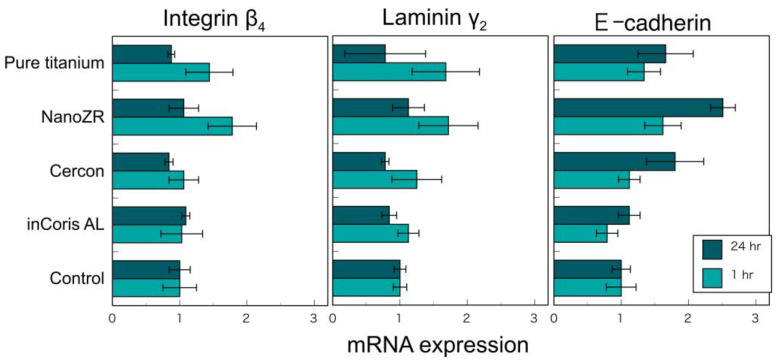
m-RNA expressions of integrin β_4_, laminin γ_2_, and E-cadherin in Ca9-22 cells on pure titanium, NanoZR, Cercon^®^ (3Y-HA zirconia), inCoris AL^®^ (alumina), and control polystyrene at 1 and 24 h. Adopted from Ref. [69].

**Figure 23 materials-14-04879-f023:**
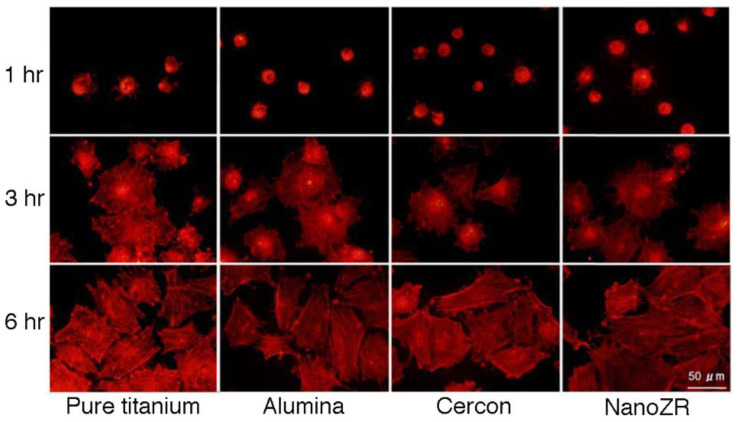
Analysis of actin cytoskeleton of MC3T3-E1 on the rough surface of pure titanium, alumina, cercon^®^ (3Y-HA zirconia), and NanoZR at 1, 3, and 6 h after incubation. Reprinted with permission from Ref. [77]. 2009 Daisuke Ymamashita.

**Table 1 materials-14-04879-t001:** Powder characteristics and typical properties of sintered body of zirconia (provided by Tosoh^®^ Corp. [28,29,31]).

		Basic Grades	S Grades	High Translucent Grades		
		TZ-3YB-E^®^	TZ-3YSB-E^®^	Zpex^®^	Zpex 4^®^	Zpex Smile^®^
Powder characteristics						
Y_2_O_3_	mol%	3	3	3	3.9	5.3
	wt%	5.2 ± 0.5	5.2 ± 0.5	5.35 ± 0.2	6.9	9.35
Al_2_O_3_ (wt%)		0.25	0.25	0.05	0.05	0.05
Actual particle size (nm)		40	90	40	90	90
Crystallite size(nm)		27	36	36	36	36
density (g/cm^3^)		1.1	1.2	1.2	1.25	1.2
Specific surface area (m^2^/g)		16	7 ± 2	13 ± 2	10	10
Typical properties of sintered body						
Green density (g/cm^3^)		3.04	3.14	3.22	3.28	3.27
Sintered temperature (°C)		1350	1450	1400	1500	1450
Sintered density (g/cm^3^)		6.06	6.05	6.08	6.07	6.04
3-P Bending strength (MPa)		1200	1200	1100	1100	600
Fracture toughness (MPa·m^0.5^)		5	5	5	-	2.4
Hardness (Hv10)		1250	1250	1250	1250	1250
Transmittance (%)		17	35	41	45	49

**Table 2 materials-14-04879-t002:** Classification of dental zirconia.

(1) Monochromic and uniform composition		
**Group**	**Name**	**Manufacturer**
ATZ	ATZ Bio-HIP	Metoxit AG (Switzerland)
NanoZR	Nanozirconia	Yamakin Co. Ltd. (Japan)
HIP 3Y-HA	TZP-A Bio-HIP	Metoxit AG (Switzerland)
3Y-HA	Cercon base	Dentsply Sirona (USA)
	inCoris Zi	Dentsply Sirona (USA)
	Vita YZ T	Vita Zahnfabrik H. Rauter GmbH & Co. KG (Germany)
	Aadva ST	GC Corp. (Japan)
	ceramill zi	Amann Girrbach AG (Austria)
	DD Bio Z	Dental Direkt GmbH (Germany)
	Z-CAD HD	Metoxit AG (Switzerland)
	Copran Zri	Whitepeaks Dental Solutions GmbH & Co. KG (Germany)
	dima Mill Zirconia ST	Kulzer GmbH (Germany)
3Y	Cercon ht	Dentsply Sirona (USA)
	Vita YZ HT	Vita Zahnfabrik H. Rauter GmbH & Co. KG (Germany)
	Aadva EI	GC Corp. (Japan)
	ceramill zolid	Amann Girrbach AG (Austria)
	inCoris TZI	Dentsply Sirona (USA)
	DD Bio ZX^2^	Dental Direkt GmbH (Germany)
	Z-CAD HTL	Metoxit AG (Switzerland)
	IPS e.max ZirCAD MO/LT	Ivoclar Vivadent AG (Lichtenstein)
4Y	Katana Zirconia HT	Kuraray Noritake Dental Inc. (Japan)
	ceramill zolid HT+	Amann Girrbach AG (Austria)
	DDcube ONE	Dental Direkt GmbH (Germany)
	Z-CAD One4All	Metoxit AG (Switzerland)
	CopraSupreme	Whitepeaks Dental Solutions GmbH & Co. KG (Germany)
	IPS e.max ZirCAD MT	Ivoclar Vivadent AG (Lichtenstein)
	Vita YZ ST	Vita Zahnfabrik H. Rauter GmbH & Co. KG (Germany)
5Y	Aadva NT	GC Corp. (Japan)
	Prettau Anterior	Zirkonzahn GmbH (Italy)
	DD cube X^2^	Dental Direkt GmbH (Germany)
	Vita YZ XT	Vita Zahnfabrik H. Rauter GmbH & Co. KG (Germany)
	Z-CAD Smile	Metoxit AG (Switzerland)
	CopraSmile	Whitepeaks Dental Solutions GmbH & Co. KG (Germany)
	Ceramill zolid fx	Amann Girrbach AG (Austria)
	Cercon xt	Dentsply Sirona (USA)
6Y	Katana Zirconia UT	Kuraray Noritake Dental Inc. (Japan)
(2) Polychromic multilayer and uniform composition		
**Group**	**Name**	**Manufacturer**
M3Y	Dima Mill Zirconia ML	Kulzer GmbH (Germany)
	Nacera Pearl Multi-Shade	Doceram Medical Ceramics GmbH (Germany)
	Prettau 2 Dispersive	Zirkonzahn GmbH (Italy)
M4Y	Katana Zirconia ML	Kuraray Noritake Dental Inc. (Japan)
	Z-CAD One4All Multi	Metoxit AG (Switzerland)
	DDcube ONE ML	Dental Direkt GmbH (Germany)
	Vita YZ ST Multicolor	Vita Zahnfabrik H. Rauter GmbH & Co. KG (Germany)
	Ceramill zolid gen-x	Amann Girrbach AG (Austria)
	CopraSupreme Symphony	Whitepeaks Dental Solutions GmbH & Co. KG (Germany)
	Shofu Block Zr Lucent CEREC	Shofu Inc./Adamant Namiki (Japan)
M5Y	Katana Zirconia Block STML	Kuraray Noritake Dental Inc. (Japan)
	Z-CAD Smile Multi	Metoxit AG (Switzerland)
	DD cube X^2^ ML	Dental Direkt GmbH (Germany)
	Ceramill zolid fx multilayer	Amann Girrbach AG (Austria)
	Cercon xt ML	Dentsply Sirona (USA)
	CopraSmile Symphony	Whitepeaks Dental Solutions GmbH & Co. KG (Germany)
	Vita YZ XT Multicolor	Vita Zahnfabrik H. Rauter GmbH & Co. KG (Germany)
	Prettau 4 Anterior Dispersive	Zirkonzahn GmbH (Italy)
	Lucent FA	Shofu Inc./Adamant Namiki (Japan)
M6Y	Katana Zirconia UTML	Kuraray Noritake Dental Inc. (Japan)
	Nacera Pearl Q^3^ Multi-Shade	Doceram Medical Ceramics GmbH (Germany)
(3) Polychromic multilayer and hybrid composition		
**Group**	**Name**	**Manufacturer**
M3Y-5Y	IPS e.max ZirCAD Prime	Ivoclar Vivadent AG (Lichtenstein)
	Prettau 3 Dispersive	Zirkonzahn GmbH (Italy)
	Tanaka Enamel ZR Multi 5	ATD Japan Co., Ltd. (Japan)
	Zivino	Yoshida Dental Co., Ltd./Adamant Namiki (Japan)
	Lucent Supra	Shofu Inc./Adamant Namiki (Japan)
M3Y-4Y	Sakura Zirconia	Straumann Japan/Adamant Namiki (Japan)
M4Y-5Y	CopraSupreme Hyperion	Whitepeaks Dental Solutions GmbH & Co. KG (Germany)
	IPS e.max ZirCAD MT Multi	Ivoclar Vivadent AG (Lichtenstein)

**Table 3 materials-14-04879-t003:** Commercially available zirconia implant system.

Manufacturer	Brand Name	Material	Design	Surface Treatment
CeraRoot SL (Barcelona, Spain)	CeraRoot	3Y-HA (Toso TZ-3YB-E)	One-piece	Acid etching (ICE surface)
Z-systems AG (Stuttgart, Germany)	Zirkolith Z5m	TZP-A Bio-HIP (HIP TZP)	One-piece	Sandblasting + Laser irradiation
	Zirkolith Z5c		Two-piece (cementation)	
	Zirkolith Z5BL		Two-piece (ceramic screw)	
METOXIT dental (Thayngen, Switzerland)	TZP-A Bio-HIP	Ziraldent HIP TZP	Two-piece	Sandblasting + Zirconia-slurry coating and firing (ZircaPore)
	ATZ Bio-HIP	Ziraldent HIP ATZ		
bredent medical (Senden, Germany)	whiteSKY Tissue Line	Y-TZP	One-piece	Sandblasting
Ziterion (Uffenheim, Germany)	Zit-vario	Y-TZP	Two-piece	Sandblasting
Dentalpoint AG (Zurich, Switzerland)	ZERAMEX T	HIP TZP	Two-piece (carbon fiber-reinforced polymer screw)	Sandblasting + Acid etching (Zerafil)
	ZERAMEX P	ATZ		
	ZERAMEX XT			
Incermed (Lausanne, Switzerland)	Sigma	HIP TZP	Two-piece	Sandblasting
AXIS biodent/Camlog (Les Bois, Switzerland)	Hexalobe	Y-TZP + PEEK	Two-piece	CIM (Ceramic Injection Molding)
	Monobloc	Y-TZP	One-piece	
Straumann (Basel, Switzerland)	PURE Ceramic Implant	HIP TZP	One-piece	Sandblasting + Acid etching (ZLA)
	SNOW Ceramic Implant (Z-systems AG)		Two-piece (Ti or ceramic screw)	Sandblasting + Laser irradiation
Nobel Biocare	Nobel Pearl (Dentalpoint AG)	ATZ	Two-piece (carbon fiber-reinforced polymer screw)	Sandblasting + Acid etching (Zerafil)
VITA Zahnfabrik	Ceramic Implant	Y-TZP	One-piece	Sandblasting + Acid etching + Heat treatment

## Data Availability

This is the review paper with minor amounts of unpublished results of the author. These data are stored by the author, not available publicity.

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
