# Peer review of "Classification and Properties of Dental Zirconia as Implant Fixtures and Superstructures"

_materials, 2021, doi:10.3390/ma14174879_

Round 1
Reviewer 1 Report
With all due respect I will graciously reject this review because already exist other reviews on this topic.
Here are:
https://www.sciencedirect.com/science/article/pii/S2238785419300419
Nistor L, Grădinaru M, Rîcă R, et al. Zirconia Use in Dentistry - Manufacturing and Properties. Curr Health Sci J. 2019;45(1):28-35. doi:10.12865/CHSJ.45.01.03
Laumbacher H, Strasser T, Knüttel H, Rosentritt M. Long-term clinical performance and complications of zirconia-based tooth- and implant-supported fixed prosthodontic restorations: A summary of systematic reviews. J Dent. 2021 Jun 11;111:103723. doi: 10.1016/j.jdent.2021.103723. Epub ahead of print. PMID: 34119611.
Kunrath MF, Gupta S, Lorusso F, Scarano A, Noumbissi S. Oral Tissue Interactions and Cellular Response to Zirconia Implant-Prosthetic Components: A Critical Review. Materials (Basel). 2021 May 25;14(11):2825. doi: 10.3390/ma14112825. PMID: 34070589; PMCID: PMC8198172.
Comisso I, Arias-Herrera S, Gupta S. Zirconium dioxide implants as an alternative to titanium: A systematic review. J Clin Exp Dent. 2021 May 1;13(5):e511-e519. doi: 10.4317/jced.58063. PMID: 33981400; PMCID: PMC8106941.
Author Response
Thank you for your comment. However, the point that "other reviews already exists" is not correct. For example, in the reviews you suggested,
- The review by Grech et al. is just a literature review about 3Y-TZP and ATZ. All the contents are reference citations without tables and figures.
- The review by Nistor et al. is just a description of the zirconia prosthesis. Moreover, all the contents are reference citations. Reference No. 41 is my paper.
- The review by Laumbacher et al. is a summary of the article search for clinical follow-up of zirconia prostheses. It is not included in my manuscript.
- The review by Kunrath et al. only deals with the biological properties of zirconia. Moreover, all the contents are reference citations. Reference No. 94 is my co-authored paper.
- The review by Comisso et al. only compares the success rates of zirconia implants and titanium implants. It is not included in my manuscript.
Although I know there are many reviews that are a bit similar, a feature of my review is that it deals with both zirconia superstructures and implants. Also, in my review, I employed my own data as much as possible to display figures. I tried not to make just a literature review. Please understand this difference.
Reviewer 2 Report
The manuscript submitted to Materials entitled “Classification and Properties of Dental Zirconia as Implant Fixtures and Superstructures ” is an original review article which aim to summarize classification, properties, and use of zirconia in dentistry. On my opinion the article is interesting, well written, with good English. Anyway, there are some minor issues to address.
English language: Minor corrections needed.
Abstract: Please structure the abstract to attract the reader's attention.
Introduction: My main suggestion is to include a brief sentence on osseointegration and factors that can affect it: <<Osseointegration has been defined as a direct and functional connection between bone and an artificial implant. Both macroscopic and microscopic characteristics of dental implants could influence the success of these procedures [doi:10.23812/20-96-L-53]>>.
Conclusions: Please insert a conclusion section.
Figures: Please improve quality and resolution.
Abbreviations: Insert a summary of abbreviations used in the text prior to “Reference” section. Other sections have been properly prepared. After making the indicated changes, the article will be suitable for publication. Thanks for the opportunity to review this manuscript.
Author Response
- “Abstract” was arranged with addition of term “osseointegration”, and some sentences were erased.
- “Introduction” was arranged with addition of term “osseointegration” and its related 5 references were added.
- “Conclusion” section was added in the last part of this review.
- I increased the font size in figures 1, 6, 7, 11, and 14 by one and stopped using shadow characters in Figure 7. Furthermore, the resolution of these figures was increased. The horizontal size of Figure 11 was increased. The resolution of figures 8, 9, 12, 13, 14, 16, 20, 21, and 22 was increased.
- “Abbreviation” section was added in prior to the reference section.
Reviewer 3 Report
The author presents a review of interest in the field of dental materials.
Maybe a conclusion would be appropriate. The bibliography can be improved with several articles.
Author Response
I added “Conclusions” session in the last part of my manuscript.
Total 12 references as possible modern were added and the contents were revised.
Round 2
Reviewer 1 Report
Dear author,
Thank you for submitting an improved version of the paper. If Editors consider that your paper is appropriate, then I have nothing further to add.
From my perspective, your paper does not bring anything new for scientific literature.
Thank you.